# Assessment of Cytokine-Induced Neutrophil Chemoattractants as Biomarkers for Prediction of Pulmonary Toxicity of Nanomaterials

**DOI:** 10.3390/nano10081563

**Published:** 2020-08-09

**Authors:** Taisuke Tomonaga, Hiroto Izumi, Takako Oyabu, Byeong-Woo Lee, Masaru Kubo, Manabu Shimada, Shingo Noguchi, Chinatsu Nishida, Kazuhiro Yatera, Yasuo Morimoto

**Affiliations:** 1Institute of Industrial Ecological Sciences, University of Occupational and Environmental Health, 1-1 Iseigaoka, Yahata-nishi-ku, Kitakyushu, Fukuoka 807-8555, Japan; h-izumi@med.uoeh-u.ac.jp (H.I.); toyabu@med.uoeh-u.ac.jp (T.O.); leebw401@med.uoeh-u.ac.jp (B.-W.L.); yasuom@med.uoeh-u.ac.jp (Y.M.); 2Department of Chemical Engineering, Graduate School of Engineering, Hiroshima University, 4-1 Kagamiyama 1-chome, Higashi-Hiroshima-shi, Hiroshima 739-8527, Japan; mkubo@hiroshima-u.ac.jp (M.K.); smd@hiroshima-u.ac.jp (M.S.); 3Department of Respiratory Medicine, University of Occupational and Environmental Health, 1-1 Iseigaoka, Yahata-nishi-ku, Kitakyushu, Fukuoka 807-8555, Japan; sn0920@med.uoeh-u.ac.jp (S.N.); c-nishi@med.uoeh-u.ac.jp (C.N.); yatera@med.uoeh-u.ac.jp (K.Y.)

**Keywords:** cytokine-induced neutrophil chemoattractant, nanomaterial, pulmonary toxicity, biomarker, rat, CINC, intratracheal instillation, inhalation exposure

## Abstract

This work determines whether cytokine-induced neutrophil chemoattractants (CINC)-1, CINC-2 and CINC-3 can be markers for predicting high or low pulmonary toxicity of nanomaterials (NMs). We classified NMs of nickel oxide (NiO) and cerium dioxide (CeO_2_) into high toxicity and NMs of two types of titanium dioxides (TiO_2_ (P90 and rutile)) and zinc oxide (ZnO) into low toxicity, and we analyzed previous data of CINCs in bronchoalveolar lavage fluid (BALF) of rats from three days to six months after intratracheal instillation (0.2 and 1.0 mg) and inhalation exposure (0.32–10.4 mg/m^3^) of materials (NiO, CeO_2_, TiO_2_ (P90 and rutile), ZnO NMs and micron-particles of crystalline silica (SiO_2_)). The concentration of CINC-1 and CINC-2 in BALF had different increase tendency between high and low pulmonary toxicity of NMs and correlated with the other inflammatory markers in BALF. However, CINC-3 increased only slightly in a dose-dependent manner compared with CINC-1 and CINC-2. Analysis of receiver operating characteristics for the toxicity of NMs by CINC-1 and CINC-2 showed the most accuracy of discrimination of the toxicity at one week or one month after exposure and CINC-1 and CINC-2 in BALF following intratracheal instillation of SiO_2_ as a high toxicity could accurately predict the toxicity at more than one month after exposure. These data suggest that CINC-1 and CINC-2 may be useful biomarkers for the prediction of pulmonary toxicity of NMs relatively early in both intratracheal instillation and inhalation exposure.

## 1. Introduction

In recent years, materials with physicochemical properties of peculiar to nanoparticles have been used in various industrial fields, such as photocatalysts, cosmetics, sunscreens, solar panels and semiconductors. At the same time, it has been reported that nanoparticles generally cause greater pulmonary inflammation than micrometer-sized particles, raising concern about their harmful respiratory effects on workers handling nanomaterials (NMs), due to inhalation via the airway. An important aspect of the pulmonary toxicity of NMs is that it involves not only lung inflammation, but also irreversible pathologic conditions such as fibrosis and tumors in the chronic phase. Thus, it is necessary to evaluate pulmonary toxicity not only in the acute phase, but also in the chronic phase. It is difficult to perform long-term exposure studies of various NMs using animals; for this reason, it is thought that a biomarker that can predict pulmonary toxicity is necessary.

Inhaled chemicals penetrate into the pulmonary alveolar space and cytokines are released from macrophages and injured lung tissue. These cytokines, especially chemokines, promote the migration and activation of neutrophils, which is how inflammation, mainly by neutrophils, progresses. Inflammation then causes irreversible lesions, fibrosis and tumors [1,2,3]. Silica and asbestos, which are known to cause lung tumors and fibrosis, have been observed to cause persistent inflammation [4,5,6], while it has been reported that low-toxicity titanium dioxide and fullerenes induced transient inflammation [7,8].

Among chemokines, cytokine-induced neutrophil chemoattractants (CINC) are classified into the CXC chemokine family and CXCL-1 (CINC-1), CXCL-2 (CINC-3) and CXCL-3 (CINC-2αβ) are reported to be involved in neutrophil migration and activation in not only rodents, but also in humans [9,10,11,12,13].

Some reports have stated that CINC is related to various types of lung inflammation. An increase of CINC-1 and CINC-2 were observed in a rat that had pneumonia with chronic *Pseudomonas aeruginosa* infection [14]. In an intratracheal instillation in rats, lipopolysaccharide (LPS) induced CINC-1 and macrophage inflammatory protein-2 (CINC-3) production in the bronchoalveolar lavage fluid (BALF) and mRNA expression in the lung [15]. It has also been reported that lung disorder caused by diesel and silica particles is related to CINCs [16,17], suggesting that CINCs are deeply involved in lung inflammation caused by respirable dust.

We previously performed an intratracheal instillation of NiO nanoparticles and examined CINC-1, CINC-2 and CINC-3 in BALF and lung tissues, reporting that CINC-1 and CINC-2 were upregulated biomarkers accompanying persistent lung inflammation [18]. These studies suggested that CINC may be involved in lung disorders caused by NMs and could be used as a biomarker of the prediction of pulmonary toxicity. We have measured the CINCs in BALF following intratracheal instillation and inhalation exposure of NMs in each previous examination [19,20,21,22], and it is no clear whether CINCs can be markers for predicting the pulmonary toxicity of NMs

In present study, we analyzed the previous data of CINC-1, CINC-2 and CINC-3 in BALF samples obtained from intratracheal instillations and inhalation exposures using NMs of differing toxicities in order to investigate whether CINCs can be a useful marker for predicting high or low pulmonary toxicity of NMs in respiratory exposure examinations.

## 2. Materials and Methods

### 2.1. Sample (Nano)Materials

Nickel oxide (NiO, primary size 19 nm) (US3355, US Research Nanomaterials, Houston, TX, USA), two types of titanium dioxide (TiO_2_ (P90); mainly composed of anatase, primary size 14 nm (Aeroxide Evonik Degussa Corp, Nordrhein-Westfalen, Germany) and TiO_2_ (rutile); composed of rutile, primary size: 12 × 55 nm (MT-150AW, Teyca Co., Ltd., Osaka, Japan)), cerium dioxide (CeO_2_, primary size 7.8 nm) (Wako Chemical, Ltd., Osaka, Japan), zinc oxide (ZnO, primary size 35 nm) (Sigma-Aldrich Co. LLC., Tokyo, Japan) and micron-sized crystalline silica (SiO_2_, primary size 1.6 µm) (MIN-*U*-SIL^®^ 5, U.S. Silica Company, Berkeley Springs, WV, U.S) were dispersed in 0.4 mL distilled water. For high toxicity nanoparticles we used NiO and CeO_2_, and for low toxicity nanoparticles we used TiO_2_ (P90), TiO_2_ (rutile) and ZnO. SiO_2_ was used as a positive control that in previous studies has shown high pulmonary toxicity [23].

### 2.2. Animals

Male Fischer 344 rats (9–11 weeks old) used in the exposure to NiO, CeO_2_, TiO_2_ (rutile), ZnO and SiO_2_ were purchased from Charles River Laboratories International, Inc. (Kanagawa, Japan). Male Wistar Hannover rats (11 weeks old) used in the exposure to TiO_2_ (P90) were purchased from Japan SLC, Inc. (Shizuoka, Japan). The animals were kept in the Laboratory Animal Research Center of the University of Occupational and Environmental Health for 2 weeks with access to free-feeding of commercial diet and water. All procedures and animal handling were done according to the guidelines described in the Japanese Guide for the Care and Use of Laboratory Animals as approved by the Animal Care and Use Committee, University of Occupational and Environmental Health, Japan.

### 2.3. Sample Preparation of Nanoparticle Suspensions

The TiO_2_ (P90) nanoparticle suspension was prepared in the manner prescribed by Oyabu et al. [24]. Briefly, TiO_2_ (P90) powder was dispersed in deionized endotoxin-free water by sonication for 45 min at 150 W using an ultrasonic sonicator (Sonifier 250, Branson, Mo, USA) and centrifuged for 20 min at 8900× *g* (Himac CR21, Hitachi, Ltd., Tokyo, Japan). About half of the supernatant was collected. The weight concentration of the TiO_2_ in the supernatant was determined and diluted to the instillation concentration with sterile purified water.

Raw nanoparticles of NiO and TiO_2_ (rutile) were dispersed in deionized endotoxin-free water. Briefly, several repetitions of ultrasonic homogenizing (Sonifier 450, 20 kHz, 350 W, Branson Ultrasonics Corporation, Danbury, CT, USA) and centrifugation of 10,000× g for 20 min centrifugation (Himac CR21, Hitachi, Ltd., Tokyo, Japan) were performed.

Raw nanoparticles of CeO_2_ were also dispersed in deionized endotoxin-free water. The CeO_2_ in the supernatants obtained by a combination of 2 h ultrasonic homogenizing (Branson 5510 J-MT, 42 kHz 180 W, Branson Ultrasonics Corporation, Danbury, CT, USA) and 20,000× g for 30 min centrifugation (Hitachi Koki Co., Ltd., CF16RX2) were in stable suspension without any agglomerated coarse particles larger than 1 µm. The detailed procedures with NiO, TiO_2_ (rutile) and CeO_2_ are explained in our previous report [25].

Commercial ZnO nanoparticle dispersion was diluted to 10 mg/mL with deionized endotoxin-free water and was well-homogenized by 2 h ultrasonic homogenizing (Branson 5510 J-MT, 42 kHz 180 W).

### 2.4. Intratracheal Instillation

The NiO, TiO_2_ (P90), TiO_2_ (rutile), CeO_2_, ZnO and SiO_2_ were suspended in 0.4 mL distilled water. A 0.2 mg dose (low dose) or 1 mg dose (high dose) was administered to rats (12 weeks old) in a single intratracheal instillation. Each of the negative control groups received dispersion mediums which were used in preparation of NM suspensions. Animals were dissected at 3 days, 1 week, 1 month, 3 months and 6 months after the instillation.

### 2.5. Inhalation Exposure

The setup used here has been described in detail in a previous paper [25,26]. Briefly, an aerosol generation system consisting of a pressurized nebulizer (Nanomaster, JSR Corp., Tokyo, Japan) and a drying section was connected to the whole-body exposure chamber in a rat cage. Inhalation exposure was performed by supplying two concentrations of aerosol particles of NiO, CeO_2_, TiO_2_ (rutile) and ZnO. The nanoparticles of the suspension were diluted with water, set in a nebulizer and sprayed with compressed air at a flow rate of 40 L/min. As a drying process, water droplets were continuously passed through a heated (150 °C) tube to remove the water. After the drying process, air containing bipolar ions supplied by an ionizer (SJ-M, Keyence Corp., Tokyo, Japan) was introduced at the same time as the aerosol flow at a flow rate of 10 L/min to neutralize the aerosol particles and reduce particle wall loss in tubes due to electrostatic forces. Clean air was added to the resulting aerosol stream to set the total airflow rate to 100 L/min, and the aerosol was supplied to the exposure chamber for 6 h each day of the 4-week inhalation test. The size and concentration of aerosol particles inside and outside the exposure chamber were analyzed by a particle size spectrometer (model 1000XP WPS, MSP Corp., Shoreview, MN, USA) built for inline monitoring. For comparison, aerosol particles were sampled with an electrostatic precipitator for off-line analysis using a field emission scanning electron microscope (FE-SEM; S-5200, Hitachi High Technologies Corp., Tokyo, Japan). In addition, the mass concentration of the aerosol in the chamber was determined by a gravimetric method in which the aerosol flowed through a fibrous filter and the weight of the collected particles was measured.

Each aerosol sample showed a very stable particle size distribution over 6 h each day of the entire inhalation study. The particle size distribution showed two peaks of about 30 and 100 nm for the NiO aerosol and of about 30 and 200 nm for the TiO_2_ (rutile) aerosol. This was due to the Coulomb explosion [26]. The particle size distributions of the CeO_2_ and ZnO aerosols peaked at about 90 nm and 120 nm, respectively. The average mass concentrations of the aerosols measured daily for the 4 weeks were as follows: The low and high concentrations of NiO were 0.32 ± 0.07 and 1.65 ± 0.20 mg/m^3^, respectively. The low and high concentrations of CeO_2_ were 2.09 ± 0.29 and 10.2 ± 1.38 mg/m^3^, respectively. The low and high concentrations of TiO_2_ (rutile) were 0.50 ± 0.26 and 1.84 ± 0.74 mg/m^3^, respectively. The low and high concentrations of ZnO were 2.11 ± 0.45 and 10.4 ± 1.39 mg/m^3^, respectively.

In order to calculate the amount of lung burden in rat lung after inhalation exposure, we used a multiple-path particle dosimetry model 2 (MPPD ver2.11) model [27] as an exposure condition (Table 1).

### 2.6. Estimation of Amount in Human Exposure Corresponding to the Intratracheal Instillation Dose in Rat

We used the following formula (1) to estimate what amount in human exposure corresponds to the intratracheal instillation dose (1 mg) in rat [28]. Assuming that inhaled chemicals would be deposited at the same rate (particle deposition efficiency 0.1, amount of deposited material/1 g of lung weight) in rats and humans and using the condition in Table 1, the estimated exposure time of nanomaterials in human (exposure concentration: 3 mg/m^3^) was calculated.
(Deposited mass of particles) = (exposure concentration of particles)               × (tidal volume) × (breathing frequency)                  × (exposure hours in day) × (days of exposure)           × (particle deposition fraction)(1)

### 2.7. Animals Following Intratracheal Instillation and Inhalation

In the exposure to NiO, TiO_2_ (P90), TiO_2_ (rutile), CeO_2_, ZnO, SiO_2_ and the control, there were 10 rats in each group, divided into two subgroups of five animals in each low dose and high dose group at each time course. In each of the first subgroups, five rats provided bronchoalveolar lavage at each time course. The lungs were inflated with 20 mL of saline at a water pressure of 20 cm and BALF was collected from the entire lung in two or three portions. A sample of 15–18 mL of BALF was collected in a collection tube by free fall. In the second subgroup, the lungs were divided into right and left lungs, and histopathological evaluation was performed on the left lung, which was inflated and fixed with 10% formaldehyde.

### 2.8. Analysis of Inflammatory Cells in BALF with Cytospin

The obtained BALF was centrifuged at 400 g at 4 °C for 15 min, and the supernatant was transferred to a new tube and frozen for measuring the cytokines. The pellets were washed by suspension with polymorphonuclear leukocyte (PMN) Buffer (137.9-mM NaCl, 2.7-mM KCl, 8.2-mM Na_2_HPO_4_, 1.5-mM KH_2_PO_4_, 5.6-mM C_6_H_12_O_6_) and centrifuged at 400 g at 4 °C for 15 min. After the supernatant was removed, the pellets were resuspended with 1 mL of PMN Buffer. The number of cells in the BALF was counted by Celltac (Nihon Kohden Corp., Tokyo, Japan), and the cells were splashed on a slide glass using cytospin. After the cells were fixed and stained with Diff-Quik (Sysmex Corp., Hyogo, Japan), the number of neutrophils and alveolar macrophages were counted by microscopic observation.

### 2.9. Measurement of Chemokines, Lactate Dehydrogenase and Heme Oxigenase-1 in BALF

The concentrations of rat CINC-1 and CINC-2 in the BALF samples in all of the examinations were measured by ELISA kits, #RCN100 and #RCN200 (R&D Systems, Minneapolis, MN, USA), respectively. The concentrations of rat CINC-3 in the BALF samples after exposure to NiO and TiO_2_ (P90) was measured by ELISA kits #RCN300 (R&D Systems, Minneapolis, MN, USA). The CINC-1 and CINC-2 values of the rats exposed to high pulmonary toxicity SiO_2_ were used for assessing the ability to discriminate the toxicity of the inhaled chemical. The concentrations of rat heme oxygenase (HO)-1 in the BALF of NiO, TiO_2_ (P90), TiO_2_ (rutile), CeO_2_ and ZnO-exposed rats were measured by an ELISA kit, ADI-EKS-810A (Enzo Life Sciences, Farmingdale, NY, USA). The activity of released lactate dehydrogenase (LDH) in the BALF of NiO, TiO_2_ (rutile), CeO_2_, and ZnO-exposed rats was measured by a Cytotoxicity Detection Kit^PLUS^ (LDH) (Roche Diagnostics GmbH, Mannheim, Germany). All measurements were performed according to the manufacturers’ instructions.

### 2.10. Histopathology

The obtained lung tissue, which was inflated and fixed with 10% formaldehyde or 4% paraformaldehyde under a pressure of 25 cm water, was embedded in paraffin and 5-mm-thick sections were cut from the lobe, then stained with hematoxylin and eosin. Inflammation scores in the histopathological findings of the lung as none (0), minimal (0.5), mild (1), moderate (2) or severe inflammation (3) were analyzed for the correlation with CINC-1 and CINC-2 after intratracheal instillation and inhalation exposure of NiO, CeO_2_, TiO_2_ (rutile), ZnO and each negative control.

### 2.11. Statistical Analysis

Analysis of variance and Dunnett’s test were applied where appropriate to determine individual differences using a computer statistical package (SPSS, SPSS, Inc., Chicago, IL, USA). Construct validity was measured using Spearman’s rank correlation coefficients between the concentration of CINC-1 or CINC-2, neutrophil counts, total cell counts, concentration of HO-1, activity of released LDH and inflammation scores in the histopathological findings of the lung. We designated the toxicity of the exposure NMs as high or low according to the values of the CINC-1 and CINC-2 concentration in each sample and analyzed with SPSS software the sensitivity and specificity for high toxicity at each time point to create the receiver operating characteristics (ROC) curves and areas under the curves (AUC)s. The cutoff values for predicting pulmonary toxicity were obtained from the ROC analysis.

## 3. Result

### 3.1. Chemical Characterization of Materials

In the present study, we used nanoparticles of NiO, two types of TiO_2_ (P90) and TiO_2_ (rutile), CeO_2_, ZnO and micron-sized SiO_2_. The physicochemical profiles of these samples are shown in Appendix A. The primary size and the purity of NiO were 19 nm and more than 99.5%, respectively. The primary size and the purity of TiO_2_ (P90) were 14 nm and more than 99.5%, respectively. The primary size and the purity of TiO_2_ (rutile) were 12 × 55 nm and 99.5%, respectively. The primary size and the purity of CeO_2_ were 7.8 nm and 99.9%, respectively. The primary size and the purity of ZnO were 35 nm and 99.9%, respectively. The primary size and the purity of ZnO were 35 nm and 99.9%, respectively. The primary size and the purity of SiO_2_ were 1.6 μm and 98.3%, respectively. The data of these samples have been published in previous studies [19,20,21,22]. As in our previous study [28], we defined the toxicity of the chemicals as follows: in intratracheal instillation studies, the chemicals that induced either persistent inflammation, fibrosis or tumor were defined as having high toxicity and the chemicals that did not induce any of those pathologic conditions were defined as having low toxicity. In previous studies [19,20,21,22], NiO and CeO_2_ induced persistent inflammation and TiO_2_ (P90), TiO_2_ (rutile) and ZnO induced transient inflammation. Accordingly, in the present study, NiO and CeO_2_ classified as chemicals with high toxicity and TiO_2_ (P90), TiO_2_ (rutile) and ZnO were classified as chemicals with low toxicity.

### 3.2. Cell Analysis in BALF and Pathologic Features in the Rat Lung

Appendix A shows the neutrophil counts in BALF that we previously reported [19,20,21,22,28]. Based on the previous reports [19,20,21,22,28], after intratracheal instillation of NiO and CeO_2_, we observed a persistent increase in neutrophil counts as compared with each negative control. After intratracheal instillation of TiO_2_ (P90), TiO_2_ (rutile) and ZnO, we observed a transient increase in neutrophils until 1 week after exposure. Based on the previous reports [19,20,21,22,28] after inhalation exposure, a tendency of an increase in neutrophil counts in the high NiO exposure group was observed until 1 month, and there was a significant increase in neutrophil counts in the low and high CeO_2_ exposure groups until 3 months. There was a transient increase or no increase of neutrophils in the inhalation exposure of ZnO and TiO_2_ (rutile), respectively. These results were obtained from previous reports [19,20,21,22,28].

Figure 1 shows inflammatory cells in BALF with cytospin at 6 months after the intratracheal instillation of the amount of 1 mg and at 3 months after the inhalation exposure of a high concentration of NMs. In the intratracheal instillation of NiO and CeO_2_, infiltration of neutrophils was observed in BALF with cytospin at 6 months (Figure 1A,B). On the other hand, no neutrophil infiltration was observed in the intratracheal instillation of TiO_2_ (P90), TiO_2_ (rutile) and ZnO (Figure 1C–E). In the inhalation exposure (Figure 1G–K), macrophages were observed in all the samples, and infiltration of neutrophils was observed at 3 months after exposure of CeO_2_ (Figure 1H).

Figure 2 shows the pathologic findings in the rat lung at 6 months after intratracheal instillation and at 3 months after inhalation exposure of NMs. In the intratracheal instillation of NiO and CeO_2_, there was infiltration of inflammatory cells such as neutrophils and macrophages in alveolar spaces in the lung at 6 months after exposure (Figure 2A,B). In the intratracheal instillation of TiO_2_ (P90), TiO_2_ (rutile) and ZnO, there was no infiltration of inflammatory cells or fibrosis at 6 months after exposure (Figure 2C–E). In the inhalation exposure of NiO, a slight infiltration of macrophages in the alveolar space was observed at 3 months after exposure (Figure 2G). In the inhalation exposure of CeO_2_, there was mainly infiltration of macrophages in the alveolar space at 3 months after exposure (Figure 2H). In contrast, no infiltration of inflammatory cells was observed in lung at 3 months after inhalation exposure of TiO_2_ (rutile) and ZnO (Figure 2I,J). These results were consistent with our previous studies [19,20,21,22].

### 3.3. CINC-1 and CINC-2 Concentrations in BALF

Figure 3 shows the concentrations of CINC-1 and CINC-2 in the BALF at each time point after intratracheal instillation of NiO, CeO_2_, TiO_2_ (P90), TiO_2_ (rutile) and ZnO. Although CINCs of each NMs were previously reported [19,20,21,22], in the present study we evaluated the tendency of increase of CINCs between high and low pulmonary toxicity of NMs. Increases in the concentrations of CINC-1 and CINC-2 were observed until 1 month in the groups with low dose (0.2 mg/rat) exposure to NiO and CeO_2_. In the high dose (1.0 mg/rat) exposures, the concentrations of CINC-1 and CINC-2 increased persistently and in a dose-dependent manner compared to the low dose exposures. In the exposure to TiO_2_ (P90), TiO_2_ (rutile) and ZnO, the concentrations of CINC-1 and CINC-2 increased mainly at 3 days and 1 week. Even in the high dose (1.0 mg/rat) exposure, there was a tendency of transient increase in the concentrations of CINC-1 and CINC-2. Figure 2E,F shows the concentrations of CINC-1 and CINC-2 in BALF at each time point after the inhalation of NiO, CeO_2_, TiO_2_ (rutile) and ZnO. In the NiO-high concentration group, the CINC-1 and CINC-2 increased from 3 days to 1 month. In both the low and high concentration CeO_2_ groups, increases in the concentration of CINC-1 and CINC-2 were observed until 3 months. There was no significant increase in CINC-1 or CINC-2 concentrations in the groups of ZnO and TiO_2_ (rutile) exposure, except for the ZnO-high concentration group at 3 days after inhalation.

We also examined the level of CINC-3 using BALF obtained after intratracheal instillation of NiO and TiO_2_ (P90). CINC-3 increased only slightly in a dose-dependent manner compared with CINC-1 and CINC-2 (data not shown).

### 3.4. Correlation between CINCs and Lung Disorder Related Markers Such as Oxidative Stress, Lung Injury and Inflammation Score

Figure 4 shows the correlation between the concentration of CINCs and inflammatory markers. The concentration of CINC-1 and CINC-2 in the BALF correlated well with the neutrophil count, the total cell counts, the concentration of rat HO-1 and the activity of LDH. Appendix A shows the relationship between CINCs and inflammation scores in intratracheal instillation and inhalation exposure of NiO, CeO_2_, TiO_2_ (rutile), ZnO and each negative control. CINC-1 and CINC-2 had positive correlation with inflammation score of histopathological findings in both of intratracheal instillation and inhalation exposure.

### 3.5. Assessment of the Accuracy of CINC-1 and CINC-2 for Measuring the Toxicity of Chemicals

Figure 5 shows the results of the receiver operating characteristics (ROC) for the toxicity of the chemicals by the concentration of CINC-1 and CINC-2 after the intratracheal instillation and inhalation exposures. The largest areas under the curves (AUC) using ROC curves for the toxicity of the chemicals in the intratracheal instillation were 1.000 (95% CI, 1.000–1.000) for both CINC-1 and CINC-2 at 1 week to 1 month. In the inhalation exposure, the largest AUCs were 0.945 (95% CI, 0.877–1.000) and 0.870 (95% CI, 0.749–0.991) in CINC-1 and CINC-2, respectively, at 1 month after exposure (Table 2).

### 3.6. The Prediction of Pulmonary Toxicity of Inhaled Chemicals Based on the Levels of CINC-1 and CINC-2

In addition to retrospective analysis based on our previous data, we confirmed experimentally, whether it is possible to predict the pulmonary toxicity based on the levels of CINC-1 and CINC-2. Therefore, we performed an intratracheal instillation of micron-sized SiO_2_ and examined the CINC-1 and CINC-2 in BALF. Figure 6 and Table 3 show that the concentrations of CINC-1 and CINC-2 following the intratracheal instillation of SiO_2_ particles were over the cutoff values at more than one month after exposure. Thus, we suggest that CINC-1 and CINC-2 can be markers to predict pulmonary toxicity by inhaled chemicals relatively early.

## 4. Discussion

In the present study, which was based on our previous studies [19,20,21,22,28], NiO and CeO_2_, which induced persistent lung inflammation, were classified as chemicals with high pulmonary toxicity and TiO_2_ (P90), TiO_2_ (rutile) and ZnO, which induced transient inflammation, were classified as chemicals with low toxicity. It has been reported that inhaled toxic chemicals such as silica and asbestos cause persistent inflammation, irreversible fibrosis and tumor [5,6]. Other studies have reported that NiO, which is considered to have high toxicity, caused persistent lung inflammation and irreversible change, such as fibrosis and tumor [29]. Persistent inflammation and fibrosis have been observed in intratracheal instillation of CeO_2_ in rats, as well [30]. On the other hand, mainly transient inflammation has been reported for TiO_2_ and ZnO, which are considered to have low toxicity [5,31,32]. It has been reported that long-term inhalation exposure to TiO_2_ induced no lung disorders such as fibrosis or tumor [5]. From these reports, we consider that our classifications of toxicity are similar to those in other reports.

In the intratracheal instillation in the present study, we used one milligram per rat as a high dose and 0.2 mg/rat as a low dose. In our previous intratracheal instillation studies, 0.2 mg/rat of NiO nanoparticle, which is a material with high toxicity, was the minimum dose for inducing inflammation in rat lung [18], while 1.0 mg/rat of TiO_2_ or fullerenes, which are materials with low toxicity, was the maximum dose that did not induce persistent inflammation [19,33]. The results of these studies indicate that pulmonary toxicity following intratracheal instillation can be evaluated by the dose range of 0.2–1.0 mg/rat.

We set the exposure concentration as 2–10 mg/m^3^ for each NM in the inhalation study. We calculated the amount of lung burden in the rat lung after inhalation exposure based on a MPPD2 model [27]. In the high concentration inhalation exposure, the lung burdens of TiO_2_ (rutile), ZnO, CeO_2_ and NiO were 0.28, 1.302, 1.43 and 0.25 mg/rat, respectively. In the present study, the estimated lung burden in inhalation exposure to nanoparticles may not be very different from the doses in intratracheal instillation, and we considered that the level in inhalation exposure was close to that in intratracheal instillation.

In intratracheal instillation and inhalation studies, we set the dose or concentration of exposure of NMs, avoiding overload which induced not only toxicity of NM itself, but also additional negative effect as well. As this manuscript described, from previous studies [18,24,33], the amount of one milligram per rat in intratracheal instillation was considered not to reach the overload, because, in intratracheal instillation of NMs having a low toxicity, one milligram was the maximum amount without delaying lung clearance of nanoparticles [24]. We estimated what amount in human exposure corresponds to the intratracheal instillation dose (one milligram) in rat. The exposure time per human would be 463 days, if one milligram per rat as the lung burden was converted into human exposure at a concentration of three milligrams per cubic meter. We think that one milligrams per rat as the lung burden of inhaled material by intratracheal instillation may correspond to approximately 1.8 years of inhalation exposure for humans at a concentration of three milligrams per cubic meter. It is thought that the amount of one milligram per rat did not reach the overload in intratracheal instillation and was the dose based on chronic exposure level in humans.

CINCs are one of the cytokines which belong to the CXC family. They have four isoforms, CINC-1 (CXCL1), CINC-2α (CXCL3), CINC-2β (CXCL3) and CINC-3 (CXCL2) and are known to be produced from macrophages and alveolar epithelial cells by LPS and inflammatory cytokines such as interleukin (IL)-1 and tumor necrosis factor and to encourage migration and activation of neutrophils [10,11,12,13,34]. CINCs are a counterpart of human growth-regulated oncogene product (GRO) in the IL-8 family. Although IL-8 is well-known as a chemokine in human, a strict IL-8 analog has not been identified in the rat [9,11]. Therefore, CINCs are considered to be important chemokines for the migration and activation of neutrophils in rats.

NiO and CeO_2_, which were classified as having high toxicity in the present study, caused increases of CINC-1 and CINC-2 persistently in both the intratracheal instillation and the inhalation exposure, while intratracheal instillation of TiO_2_ (P90), TiO_2_ (rutile) and ZnO, which were classified as having low toxicity in the present study, induced transient increases. A transient increase of CINC-1 and CINC-2 was observed in the inhalation of ZnO. Inhalation exposure of TiO_2_ (rutile) did not induce any increase. Our observations, including the chronic phase, suggest that the tendencies of an increase of CINC-1 and CINC-2 differed according to toxicity in both the intratracheal instillation test and the inhalation exposure. In intratracheal instillation in rats, it has been reported that bolus exposure caused transiently negative effects, even if the inhaled chemical had low toxicity [35]. Some studies have reported that ZnO and copper oxide, which have high solubility, were highly cytotoxic and induced transient lung inflammation in a rodent model [36,37], so we considered that the solubility of ZnO may have affected a transient increase of CINCs in the present study. Based on these results, we suggest that when evaluating the toxicity of a nanomaterial, it is necessary to consider that the acute phase responses may be different, depending on the exposure route and the characteristics of the chemicals.

The highest values of AUCs in the intratracheal instillation and in the inhalation exposure appeared at one week to one month after exposure. The results of the ROC analysis have been interpreted as follows: AUC < 0.70, low diagnostic accuracy; AUC in the range of 0.70–0.90, moderate diagnostic accuracy; and AUC ≧ 0.90, high diagnostic accuracy [38]. Therefore, we think that the concentration of CINC-1 and CINC-2 at the subacute level could be a highly accurate marker to assess pulmonary toxicity.

In the intratracheal instillation of NMs, pulmonary toxicity is determined by the inflammation from the initial stage of exposure and whether the inflammation is persistent or disappears by the chronic phase. Some highly toxic materials, such as silica, however, do not have a pattern in which inflammation is persistent from the beginning of exposure. Intratracheal instillation studies of silica have shown that inflammation in the acute phase is minimal, but subsequent inflammation is exacerbated and persistent [39,40]. This pattern was also observed in citrate-coated silver and indium oxide nanoparticles [41,42]. In order to reflect the different inflammatory pattern of nanoparticle, we conducted an intratracheal instillation of crystalline silica (SiO_2_) to examine whether assessment of toxicity by CINC-1 and CINC-2 would be possible for inhaled chemicals with different inflammation patterns. The ability to assess pulmonary toxicity by CINC-1 and CINC-2 was then evaluated by the cutoff value obtained by ROC analysis. In the results, the high toxicity of silica, known to be a highly toxic chemical, could be determined by both CINC-1 and CINC-2 one month or more after the exposure. This suggested that even if inhaled chemicals had different inflammation patterns, both CINC-1 and CINC-2 could identify their pulmonary toxicity relatively early and could serve as biomarkers for the prediction of pulmonary toxicity.

In this study, both CINC-1 and CINC-2 had a positive correlation with total cell counts, neutrophil counts, LDH as a tissue injury marker and HO-1 as an oxidative stress marker in BALF. There are some reports that CINC is related to lung disorder in exposure to inhaled chemicals [43,44]. Exposure to tobacco particles in rats showed elevated CINC-1 and MPO activity [43]. When rats were exposed to silver nanoparticles coated with polyvinyl–pyrrolidone, the tendency for an increase of CINC-1 was similar to LDH [44]. Based on the results of the correlation of CINCs with other inflammatory markers in this study, it is thought that there is a series of steps following the release of CINC by NMs, the promotion of migration of neutrophils and the progression of lung injury.

Infiltration of neutrophils and macrophages in the lung exposed to NMs were also observed in the case of bacterial infection due to contamination. In our previous studies, nanoparticles of NiO or CeO_2_ induced infiltration of neutrophils and macrophages [19,21]. These pathologic findings corresponded to these reported in other studies [29,37]. Moreover, we could not detect bacteria in the BALF and pathologic samples. Therefore, it is considered that these inflammations observed in lung exposed to NMs classified to sterile ones in the present study. As for a mechanism of the sterile inflammation, it is reported that chemokines such as KC (CINC-1: CXCL1) induced infiltration of neutrophils through IL-1β in Nalp3-/- mice exposed to asbestos [45]. On the other hand, it has been reported that damage-associated molecular patterns (DAMPs), which are secondarily released from tissue destruction and cell death due to inflammation, lead to the expression of CXCL1(CINC-1) through inflammatory cytokines such as IL-1α [46,47,48]. Considering the increase in released LDH [19,21,22] and in CINCs in the present study, it is suggested that the lung inflammation caused by NMs may be related to sterile inflammation through DAMP-sensing receptors.

In this study, we examined the usefulness of CINC-3 to evaluate pulmonary toxicity using BALF obtained by intratracheal instillation of NiO and TiO_2_ (P90). However, CINC-3 increased only slightly in a dose-dependent manner compared with CINC-1 and CINC-2 and had a poor correlation with other inflammatory markers, including neutrophils in BALF (data not shown). Jeong et al. performed an intratracheal instillation of cobalt compound nanoparticles in rats and neutrophils were elevated in BALF, but not accompanied by an increase of CINC-3 [49]. We also reported that, in intratracheal instillation studies of nickel oxide nanoparticles with different aggregate diameters, persistent increases of CINC-3 were observed in severe inflammation caused by a high dose (1.0 mg), and there was no increase in the mild inflammation caused by a low dose (0.2 mg) [18,50]. These reports suggest that CINC-3, unlike CINC-1 and CINC-2, has an enhancing effect rather than an inducing effect of inflammation. In the present study, the usefulness of CINC-3 was considered to be low, because it is necessary to detect early inflammation as a condition for predicting pulmonary toxicity early.

As a conventional biomarker, neutrophil counts for evaluation of pulmonary inflammation in BALF have been reported [30,36,39]. In the present study, neutrophil counts in the BALF of rats exposed to ZnO or TiO_2_ with low toxicity persisted for one week after intratracheal instillation. If neutrophil counts were used for the prediction of pulmonary toxicity of NMs at one week after intratracheal instillation, neutrophil counts could not sufficiently reflect them. On the other hand, CINC-1 and CINC-2 can reflect the pulmonary toxicity of NMs compared to neutrophil counts, because there was a tendency of decrease of CINC-1 and CINC-2 concentration in BALF already at one week after intratracheal instillation. Moreover, since CINC-1 and CINC-2 can indicate the pulmonary toxicity of NMs at one week and one month after exposure, CINCs may be useful as biomarkers for long-term prediction of pulmonary toxicity.

## 5. Conclusions

In this study, we focused on CINCs as related to the process of neutrophilic inflammation and examined their usefulness as a biomarker for the prediction of pulmonary toxicity of nanoparticles. The results of the measurement of CINC-1 and CINC-2 (but not CINC-3) concentration reflected pulmonary toxicity after both intratracheal instillation and inhalation exposure. CINC-1 and CINC-2 correlated with inflammatory cells, LDH as a tissue injury marker and HO-1 as an oxidative stress marker. In exposure to SiO_2_, which has high pulmonary toxicity and progresses to lung disorder in the chronic phase, CINC-1 and CINC-2 were able to predict the toxicity relatively early. Taken together, we suggest that CINC-1 and CINC-2 can be useful biomarkers for the prediction of pulmonary toxicity of NMs in both intratracheal instillation and inhalation exposure.

## Figures and Tables

**Figure 1 nanomaterials-10-01563-f001:**
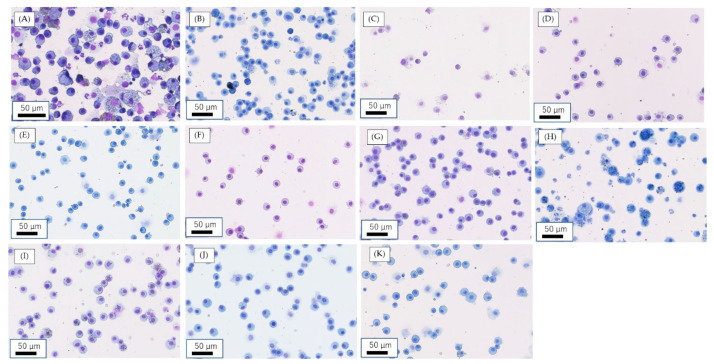
Inflammatory cells in bronchoalveolar lavage fluid (BALF) with cytospin at 6 months following intratracheal instillation and at 3 months following inhalation exposure. (**A**) Instillation of 1.0 mg NiO; (**B**) instillation of 1.0 mg CeO_2_; (**C**) instillation of 1.0 mg TiO_2_ (P90); (**D**) instillation of 1.0 mg TiO_2_ (rutile); (**E**) instillation of 1.0 mg ZnO; (**F**) negative control in intratracheal instillation (distilled water); (**G**) high concentration (1.65 ± 0.20 mg/m^3^) of NiO; (**H**) high concentration (10.2 ± 1.38 mg/m^3^) of CeO_2_; (**I**) high concentration (1.84 ± 0.74 mg/m^3^) of TiO_2_ (rutile); (**J**) high concentration (10.4 ± 1.39 mg/m^3^) of ZnO; (**K**) negative control of inhalation exposure (clean air).

**Figure 2 nanomaterials-10-01563-f002:**
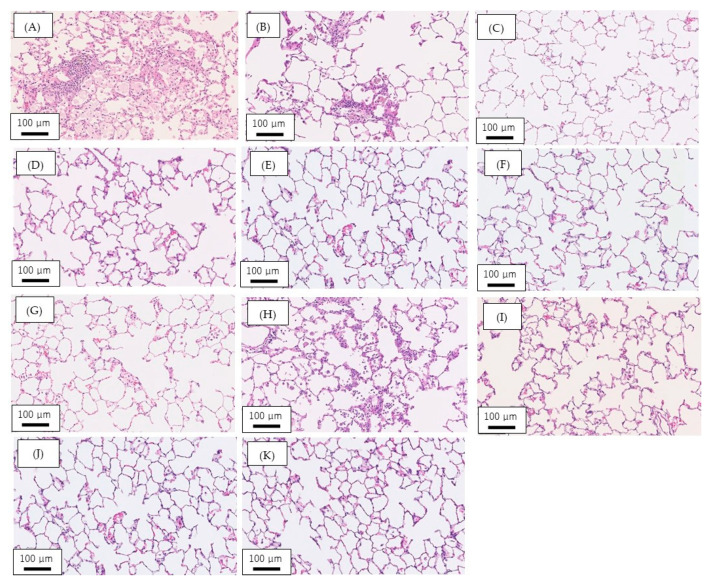
Pathologic features in the rat lung at 6 months after intratracheal instillation and at 3 months after inhalation exposure of NMs. (**A**) 1.0 mg NiO-instilled lung; (**B**) 1.0 mg CeO_2_-instilled lung; (**C**) 1.0 mg TiO_2_ (P90)-instilled lung; (**D**) 1.0 mg TiO_2_ (rutile)-instilled lung.(**E**) 1.0 mg ZnO-instilled lung; (**F**) negative control in intratracheal instillation (distilled water); (**G**) NiO-exposed lung (high concentration: 1.65 ± 0.20 mg/m^3^); (**H**) CeO_2_-exposed lung (high concentration: 10.2 ± 1.38 mg/m^3^); (**I**) TiO_2_ (rutile)-exposed lung (high concentration: 1.84 ± 0.74 mg/m^3^); (**J**) ZnO-exposed lung (high concentration: 10.4 ± 1.39 mg/m^3^); (**K**) negative control in inhalation exposure (clean air).

**Figure 3 nanomaterials-10-01563-f003:**
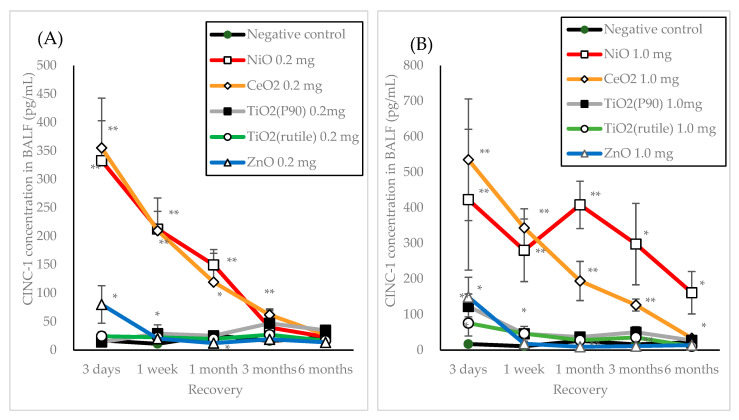
Cytokine-induced neutrophil chemoattractants (CINC)-1 and CINC-2 concentrations in BALF at each time point after intratracheal instillation and inhalation exposure of nanomaterials (NMs). (**A**) CINC-1 in low dose (0.2 mg/rat) of intratracheal instillation; (**B**) CINC-1 in high dose (1.0 mg/rat) of intratracheal instillation; (**C**) CINC-2 in low dose (0.2 mg/rat) of intratracheal instillation; (**D**) CINC-2 in high dose (1.0 mg/rat) of intratracheal instillation; (**E**) CINC-1 in inhalation exposure; (**F**) CINC-2 in inhalation exposure. Asterisks indicate significant differences compared with each control (analysis of variance and Dunnett’s test) (**p* < 0.05, ***p* < 0.01). Data point at each observation time represent average of 5 rats and the error bars show the standard deviations.

**Figure 4 nanomaterials-10-01563-f004:**
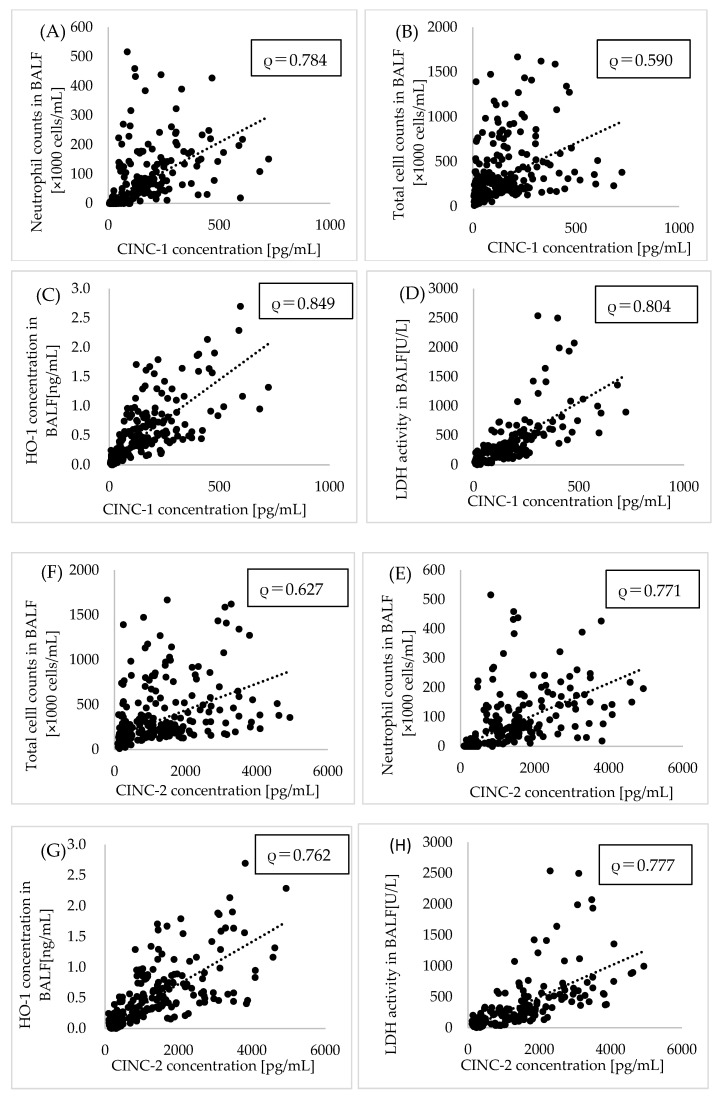
Relationship between CINCs and inflammatory, oxidative stress or lung injury. (**A**) neutrophils, (**B**) total cell (**C**) rat heme oxygenase (HO)-1 and (**D**) lactate dehydrogenase (LDH) versus CINC-1 and (**E**) neutrophils, (**F**) total cell (**G**) HO-1 and (**H**) LDH versus CINC-2 in BALF after inhalation and intratracheal instillation of NMs. Values of ρ are Spearman’s rank correlation coefficient for all the data.

**Figure 5 nanomaterials-10-01563-f005:**
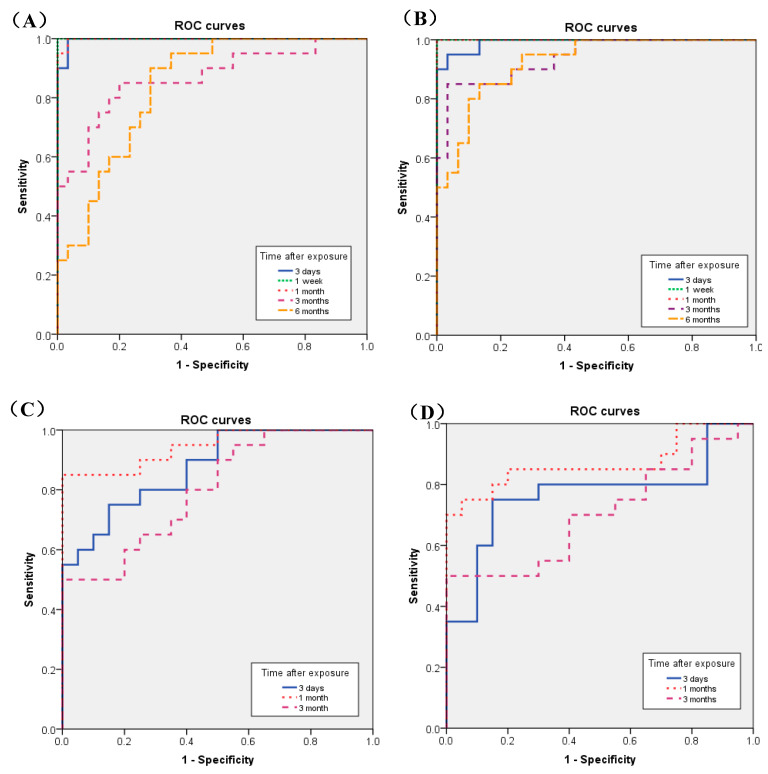
Receiver operating characteristics (ROC) for the toxicity of NMs by the concentration of CINC-1 and CINC-2. (**A**) CINC-1 in intratracheal instillation; (**B**) CINC-1 in inhalation exposure; (**C**) CINC-2 in intratracheal instillation; (**D**) CINC-2 in inhalation exposure.

**Figure 6 nanomaterials-10-01563-f006:**
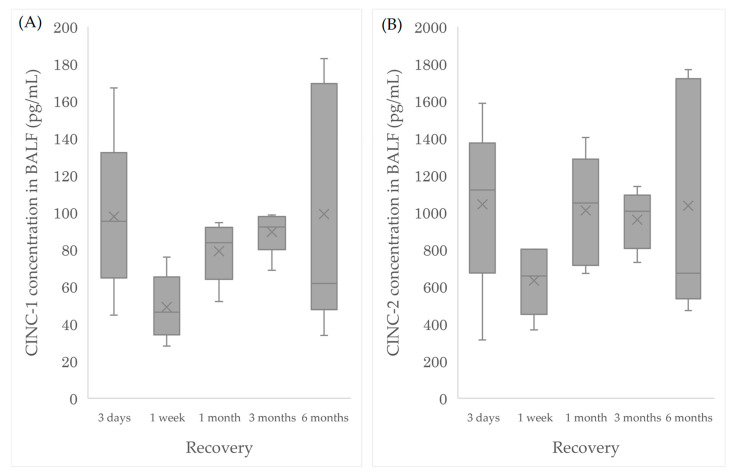
Concentration of CINC-1 and CINC-2 in BALF following intratracheal instillation of SiO_2_. (**A**) CINC-1 and (**B**) CINC-2 showed that the increasing tendencies were bimodal.

**Table 1 nanomaterials-10-01563-t001:** Calculation conditions of multiple-path particle dosimetry model 2 (MPPD2) model and estimation of exposure time in humans.

	NiO	CeO_2_	TiO_2_ (Rutile)	ZnO
Exposure concentration of nanoparticles (mg/m^3^)	1.65	10.2	1.8	10.4
Tidal volume in rat/human (mL/time)	2.1/625	–	–	–
Breathing frequency in rat/human (times/min)	102/12	–	–	–
Exposure hours per day (rat/human)	6/8	–	–	–
days of exposure (rat)	20	–	–	–

**Table 2 nanomaterials-10-01563-t002:** Results of receiver operating characteristics (ROC) analysis in intratracheal instillation and inhalation exposure of NMs.

**(A) Intratracheal Instillation**									
	**Time**	**AUC ^1^**	**(95% CI)**	***p*-Values**	**Cutoff (pg/mL)**	**Sensitivity**	**Specificity**	**PPV ^2^**	**NPV ^3^**
**CINC-1**	**3 days**	0.997	0.988–1.000	0.000	188	1.00	0.97	0.95	1.00
	**1 week**	1.000	1.000–1.000	0.000	100	1.00	1.00	1.00	1.00
	**1 month**	0.998	0.993–1.000	0.000	44	1.00	0.97	0.95	1.00
	**3 months**	0.865	0.755–0.975	0.000	44	0.85	0.80	0.74	0.89
	**6 months**	0.837	0.729–0.945	0.000	21	0.90	0.70	0.67	0.91
**CINC-2**	**3 days**	0.992	0.975–1.000	0.000	1450	0.95	0.97	0.95	0.97
	**1 week**	1.000	1.000–1.000	0.000	851	1.00	1.00	1.00	1.00
	**1 month**	1.000	1.000–1.000	0.000	569	1.00	1.00	1.00	1.00
	**3 months**	0.940	0.875–1.000	0.000	726	0.85	0.97	0.94	0.91
	**6 months**	0.923	0.853–0.994	0.000	323	0.85	0.87	0.81	0.90
**(B) Inhalation Exposure**									
	**Time**	**AUC ^1^**	**(95% CI)**	***p*-Values**	**Cutoff (pg/mL)**	**Sensitivity**	**Specificity**	**PPV ^2^**	**NPV ^3^**
**CINC-1**	**3 days**	0.875	0.771–0.979	0.000	130	0.70	0.85	0.82	0.74
	**1 month**	0.945	0.877–1.000	0.000	26	0.85	1.00	1.00	0.87
	**3 months**	0.800	0.666–0.934	0.001	18	0.80	0.60	0.67	0.75
**CINC-2**	**3 days**	0.768	0.609–0.926	0.004	312	0.80	0.70	0.73	0.78
	**1 month**	0.870	0.749–0.991	0.000	295	0.85	0.80	0.81	0.84
	**3 months**	0.705	0.539–0.871	0.027	277	0.70	0.60	0.64	0.67

^1^ AUC—area under curve. ^2^ PPV—positive predictive value; ^3^ NPV—negative predictive value.

**Table 3 nanomaterials-10-01563-t003:** Comparison of the cutoff values and the concentration of CINCs following intratracheal instillation of SiO_2_.

		3 Days	1 Week	1 Month	3 Months	6 Months
**CINC-1**	**Cutoff value (pg/mL)**	188	100	44	44	21
	**No. of above cutoff**	0/5	0/5	5/5	5/5	5/5
**CINC-2**	**Cutoff value (pg/mL)**	1450	851	569	726	323
	**No. of above cutoff**	1/5	0/5	5/5	5/5	5/5

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
