# Peer review of "Assessment of Cytokine-Induced Neutrophil Chemoattractants as Biomarkers for Prediction of Pulmonary Toxicity of Nanomaterials"

_nanomaterials, 2020, doi:10.3390/nano10081563_

Round 1
Reviewer 1 Report
The study by Tomonage et al. investigates the application of cytokine-induced neutrophil chemoattractants (CINC) as markers predicting toxicity of selected nanomaterials in rats exposed by intratracheal instillation and inhalation. The authors further correlated their finding with other parameters (neutrophil and total cell counts, LDH and HO-1 analyses) and concluded that CINC-1 and CINC-2 may serve as useful biomarkers of pulmonary toxicity of nanomaterials.
Overall, the authors should better explain the advantage of application of CINC analyses over conventional biomarkers of pulmonary inflammation. The results indicate a good correlation between CINC concentrations and neutrophil counts. It seems to be easier to assess cell numbers and avoid additional analyses of CINC.
There are numerous points in the manuscript that should be addressed.
Abstract should be revised: expressions “almost increased”, “almost no increase” should not be used; a list of tested particles and exposure times should be provided. The information that BALF was obtained from previous examinations is not important in the abstract and should be omitted.
Introduction: line 57, chemokines are related not only to lung inflammation, a better definition is needed. As CINC are reported to be typical for rats (line 58), then the results are not applicable for other animals. This again questions suitability of CINC as markers of nanoparticles exposure.
Materials and Methods, Animals, line 86: why Wistar rats were used for TiO2 (P90) exposure? The same strain for all experiments should be preferably used.
Results, section 3.3: the results reported in Fig. 2E and 2F do not correspond to description of inhalation experiment (section 2.4).
Results, section 3.4: HO-1 and LDH are not inflammatory markers, as stated in the section title.
Results, section 3.6: comparison of the data obtained for nanoparticles with those for SiO2 seems to be problematic. The particles differ chemically and by their size and it is difficult to extrapolate the data obtained for nanoparticles to SiO2 particles. Better explanation of this approach is needed. In Fig. 5, y-axis description should be corrected.
Discussion, line 302: calculation of nanoparticles concentrations should be provided in Methods section.
Discussion, line 340: the reason for conducting the experiments with SiO2 should be given in the Methods and/or Results.
Discussion, line 378: analysis of CINC-3 – the data should be commented on in the Results.
Conclusions, line 406: oxidative stress marker – HO-1, not LDH is an oxidative stress marker.
The manuscript requires language editing.
Author Response
Thank you for reviewing our manuscript entitled by “Assessment of cytokine-induced neutrophil chemoattractants as biomarkers for ranking of pulmonary toxicity of nanomaterials”. We revised the document in our manuscript according to the comments of reviewers.

Reviewer 2 Report
Review « assessment of cytokine-induced neutrophil chemoattractants as biomarkers for ranking of pulmonary toxcity of nanomaterials ».
This work is very interesting, and the data are convincing. Nevertheless, some improvements are needed.
Minor recommendation :
Figure 5 has to be translated in english.
Major recommendations :
The suspension of the nanomaterials needs to be more detailed, sonication, time, power…
The physicochemical characterizations of the NM (DLS, TEM) have to be provided.
Could the authors provide the dosimetry of the effective dose delivered after inhalation ?
Could the author justify the doses used ?
Author Response

(The authors gave the same response as above.)

Reviewer 3 Report
The authors present CINCs as biomarkers of exposure of rats to various types of nanomaterials analyzed in BALF. The rats were exposed through various administration routes and CINCs concentration was investigated in long-term conditions. The authors show that CINCs expression can be useful as biomarker of pulmonary nanotoxicity upon both, instillation and inhalation exposures. The study brings interesting and novel insights into the nanotoxicological field and opens new avenues for investigation of pulmonary toxicity of nanomaterials. The manuscript fits within the scope of the journal. Hence, I suggest its accepting after discussing on few minor issues:
What was the reason of using two types of rat experimental models?
In abstract - the authors write that Ce dioxide persistently almost increased ... what does it mean almost increased? Or is it a typo?
Fig. 5: y-axis - some weird characters are present. Need to be corrected.
Author Response

(The authors gave the same response as above.)

Reviewer 4 Report
The authors of this manuscript undertook a study into the correlation between expression of cytokines and nanoparticle linked pulmonary toxicity in a rat injury model. This study is an extension of a previous work demonstrating correlations between inflammation and nanoparticle inhalation. The authors suggest that CINC-1 and CINC-2 may serve as biomarkers for pulmonary toxicity and offer the benefit of material distinction that is not found through conventional neutrophil counts. Overall the manuscript offers little novel findings not covered in previous studies. It is difficult to find exactly what differentiates this manuscript from their previous work. While I do see value in their focus on CINC-1 and 2, the data itself does not sufficiently explore these chemokines to support publication. There are some suggestions below that may bolster and expand the work.
- The manuscript requires a broad revision in English grammar. Example in Line 394 “reflect the pulmonary toxicity of nanomaterials compared neutrophil counts”. There is the absence of the preposition “to” prior to neutrophil. A small example found throughout the manuscript.
- Table 1 is a summary of previously published studies classifying each material as high or low toxicity. This table is best moved to the supplementals and summarized in the methods section. The authors should include only novel findings in their results section.
- There is no classification system for the histological findings in figure 1. The authors suggest a differential infiltration of macrophages and neutrophils under the various nanoparticle exposures. They also suggest the absence of fibrosis in any of their models. The histology presented in these images appear to support their conclusions, however quantification of these data must be made. There are multiple assays that that can be run to confirm their conclusions. Flow cytometry or immunohistochemistry targeting specific cell surface markers allows for quantification of immune cell infiltration. Analysis of collagen deposition or fibroblast invasion would allow for quantification of fibrotic endpoints.
- Figures 2-4 present the bulk of the novel findings in this manuscript. The increased correlation between CINC-1 and 2 concentration (misspelled in X axis of figure 3) is the key finding of their work. The characterization of early and late correlation is intriguing and should be expanded. Is there specific quantifiable changes in immune cell populations that can be seen at these time points. What is the source or role of CINC-1 and 2 at these time points? These questions are the natural extension of their previous publications.
- The Y axis in figure 5 is not in English.
- It would be beneficial to include sample BAL stains to support the neutrophil counts in figure 3
Author Response

(The authors gave the same response as above.)

Round 2
Reviewer 2 Report
The authors have replied to my answers. they have to put the justification of the dose in the definitive manuscript. I agree for the publication of this article on Nanomaterials.
Author Response
Thank you for reviewing our revised manuscript entitled by “Assessment of cytokine-induced neutrophil chemoattractants as biomarkers for ranking of pulmonary toxicity of nanomaterials”. We revised the document in our manuscript according to the comments of reviewer.

Reviewer 4 Report
This is a second review of a manuscript exploring the use of CINC1 and CINC2 as early biomarkers for the ranking of pulmonary toxicity of nanomaterials. The initial review of this manuscript noted that the authors presented little novel data that was not included in the 4 previously published studies from 2015-2016 (references 25-28 in their updated manuscript). It was noted by this reviewer that the specific use of CINC1 and CINC2 as biomarkers is in fact a novel discovery that is worth exploring and expanding. As the work stands now though, this characterization is not complete and appears to be no more than an interesting corollary.
- The authors do point out effectively that CINCs are seen earlier than elevated neutrophil counts, and are well founded in arguing that this presents an intriguing method to rank toxicity of nanomaterials. The question that follows is simply weather this is a reflection of the material interaction with the immune system or is this a true measure of lung injury? The authors suggest that activated neutrophils are correlated to expression levels of CINC1 and 2, however the statement is not supported by the evidence presented. The authors report increased neutrophil number, however, activated neutrophils are marked by the expression of a transcriptional signature and specific surface markers. This is not demonstrated in this manuscript, and furthermore in response to a suggestion by this reviewer to use flow cytometry or immunohistochemistry the authors note “Although we have a great interest in examining the quantification of immune cell infiltration, the objective of our research is neither what kind of cell population play key role in pulmonary inflammation induced by nanoparticles nor whether CINCs expression is related to activation of cell population in the lung”. Given the responses by the authors it appears that their interest is specifically in the correlation of CINC protein levels to the number of neutrophils in the BAL. This is not a direct measure of lung injury and, given their claim that CINC1 and 2 are improved biomarkers for toxicity over neutrophil count, it is ineffective to base a large portion of their results on neutrophil counts in the BAL.
- The authors were asked to include a classification for histological findings to further asses the correlation between CINCs and pulmonary toxicity. The authors state in line 291 that no fibrosis is observed after 6 months of exposure and that infiltration is mainly macrophages after inhalation exposure of NiO and CeO2. While I agree that their histology presented suggests this may be the case, there is no quantification or assay to qualify this statement. The authors should differentiate between BAL and lung infiltrates. The cells found in BAL are a sampling of upper airway immune cells, but by no means a complete reflection of the lung infiltrate. Thus, to support the claim of specific infiltrate and specific histological findings, the authors must apply methodologies/assays beyond H&E.
Author Response

(The authors gave the same response as above.)
